# Awareness and Acceptance for COVID-19 Booster Dose Vaccination among Residents of Saudi Arabia: Findings of a Cross-Sectional Study

**DOI:** 10.3390/vaccines11050929

**Published:** 2023-05-03

**Authors:** Mohammad Shakil Ahmad, Tayseer Dhayfallah Almuteri, Abdulrahman Obaid Matar Alharbi, Abdullah Tawakul, Mohammed Abdulrahman Mohammed Alturiqy, Mansour Alzahrani, Shaden Bader Almutairi, Ghadah Mohammed Almutairi, Aseel Saleh Alotaibi, Nouf Sultan Almutairi, Lama Khalid Alhabdan, Waleed Khalid Z Alghuyaythat

**Affiliations:** 1Department of Family and Community Medicine, College of Medicine, Majmaah University, Majmaah 11952, Saudi Arabia; m.alzahrani@mu.edu.sa; 2Department of Obstetrics and Gynecology, College of Medicine, Majmaah University, Majmaah 11952, Saudi Arabia; td.almutairi@mu.edu.sa; 3Department of Neurology, College of Medicine, Majmaah University, Majmaah 11952, Saudi Arabia; ao.alharbi@mu.edu.sa; 4Internal Medicine Department, Faculty of Medicine, Umm Al-Qura University, Makkah 24382, Saudi Arabia; aa.tawakul@gmail.com; 5Department of Radiology, College of Medicine, Majmaah University, Majmaah 11952, Saudi Arabia; m.alturiqy@mu.edu.sa; 6College of Medicine, Majmaah University, Majmaah 11952, Saudi Arabia; shadenbadr77@gmail.com (S.B.A.); gh.almutairi7@hotmail.com (G.M.A.); iaseelalotaibi@hotmail.com (A.S.A.);

**Keywords:** COVID-19, vaccination, booster, Saudi Arabia, awareness, acceptance

## Abstract

Background: In the global effort to combat SARS CoV2 infection, adoption of the vaccination has been an essential component. The goal of this research was to determine the quality of web-based information gathered during COVID-19 and participants’ awareness and acceptance for the booster dose for COVID-19. Methods: This cross-sectional study was carried out to gauge interest in and willingness for a booster dose, as well as the satisfaction with the availability and accuracy of Internet resources. This study included 631 people from the cities of Riyadh, Al Majma’ah, Al Ghat, and Zulfi in the Riyadh Area. Chi-square and Fischer’s exact tests, with a 95% confidence interval, and a threshold of *p* < 0.05, were used to analyze the significance of associations between variables. Results: Out of 631 respondents, 347 people who reported willingness to receive the immunization were women (319, 91.9%), with only 28 (8.1%) being men. There was a statistically significant correlation between individuals who worried about booster dosage adverse effects and those who did not receive the immunization. Knowledge of the efficacy of the vaccine, confidence in the capacity of the vaccine to avert problems, and willingness to receive a third dosage were all shown to be substantially correlated (*p* < 0.001). Attitude and behavior ratings were substantially correlated with prior COVID-19 immunization status (*p* < 0.005). Conclusions: There was a significant correlation between vaccination knowledge, confidence in the capacity of the vaccine to prevent problems, and willingness to receive a third dose. Therefore, our research can help policymakers develop more precise and scientific roll-out strategies for the COVID-19 booster vaccination.

## 1. Introduction

Several diseases, from the common cold to MERS-CoV and severe acute respiratory syndrome, are caused by viruses from the Coronavirus family (SARS-CoV). As compared to the SARS-CoV, which was spread from civet cats to people in China in 2002, the MERS-CoV was spread from dromedary camels to humans in Saudi Arabia in 2012 [1]. The vaccination against SARS CoV2 has been seen to decrease the likelihood of both symptomatic and asymptomatic illness and hampers the transmission. Recent research has shown that those who have had COVID-19 but have not been vaccinated are at twice the risk of developing the disease again [2,3]. For the global effort to combat the COVID-19 pandemic, the adoption of the COVID-19 vaccine was a crucial and risky first step. Public health officials may need to address several barriers to increasing vaccination rates to combat public misperceptions about the disease, such as skepticism and stigma [4].

Since its launch on 17 December 2020, the Pfizer-BioNTech vaccine has been administered in the Kingdom of Saudi Arabia. Those in the medical field were among the first to receive vaccinations because of the high risk they face from disease transmission. The second and third waves targeted the most at-risk demographic: adults over the age of fifty. In addition, changes in Coronaviruses have resulted in the formation of novel variations such as the Delta, which poses a significant danger to herd immunity; hence, a higher number of individuals need to be inoculated to meet the threshold required to decrease the circulation of the virus [5]. Upon scientific confirmation of the necessity, the government of Saudi Arabia started providing high-risk populations with the option of receiving a third dosage of the vaccination. Those over the age of 60, those who received their second dosage more than eight months ago, and those who had chronic renal failure or had an organ transplant were all at increased risk of developing severe disease due to COVID-19 [6].

In the context of information related to the COVID-19 vaccine, reliable online resources are crucial for mitigating the effects of a pandemic. During a pandemic, it is imperative that authorities give limits and advice to the public in order to facilitate the implementation of healthy, preventative procedures. The media facilitated rapid communication and public health instructions between the Centers for Disease Control and Prevention (CDC) and the World Health Organization (WHO). Concern with COVID-19 has been amplified by the prevalence of misleading or irrelevant results in Internet searches [7,8].

Vaccine acceptance rates have been found to vary by region, according to a recent systematic review. Acceptance was low in certain countries such as Jordan (28.4%) and Kuwait (23.6%), while it was moderate in others such as Saudi Arabia (64.7%) and Russia (54.9%). The countries of eastern Asia, such as Malaysia (94.3%), Indonesia (93.3%), and China (91.3%), were shown to have the highest rates of acceptance. In order to achieve worldwide herd immunity and put a stop to the pandemic, it may be necessary to increase vaccination uptake in some nations [9]. However, there are still certain aspects that need to be addressed, for example, sociodemographic characteristics such as education level, and attitudes, political stances, and perceptions of COVID-19 that influence people’s decisions to receive the vaccine. Hence, this study was carried out with an aim to evaluate the quality of web-based information gathered during COVID-19 and the extent to which participants were made aware of the need for a booster dosage.

## 2. Materials and Methods

### 2.1. Study Design, Setting, and Participants

This COVID-19 cross-sectional online survey was conducted between 11 February and 1 March 2022 to determine the need for a booster dosage, the level of acceptability for such a measure, and the level of the quality of the accessible web-based information. As of the time of writing, 631 individuals have responded to the survey. Adult men and girls from the Riyadh Region of Saudi Arabia (including the cities of Riyadh, Al Majma’ah, Al Ghat, and Zulfi) were randomly recruited for the research.

### 2.2. Sampling Procedure and Study Tool

The information was gathered by using a self-administered survey that was linked to the database through social media. Only residents of the Riyadh area were surveyed for this data, which was broken down into three categories: (1) sociodemographic variables (such as age, gender, education level, and social status); (2) knowledge of and willingness to receive the booster dose; and (3) the credibility of health information found on the Internet.

By utilizing Google Forms, an electronic version of the survey was created and disseminated via a variety of social media platforms such as Twitter, WhatsApp, and Telegram, which were specific for Riyadh region. The responses were conceived on the spreadsheet, and the anonymity of the respondent was maintained in the questionnaires. The link was open for 20 days, and the responses obtained during that period were included in the study after excluding the incomplete responses. The duplicate entries detected by the same email account were excluded.

The survey that was used in this study was written in Arabic. The questionnaire was developed and reviewed by the public health faculty working in Majmaah University. The questionnaire was tested for reliability and validity by conducting a pilot study. In total, 50 responses were recorded in the pilot study, and the Cronbach’s alpha coefficient that was calculated was >0.7. Questions in the instrument that were included were on online health resources for knowledge on COVID-19 and readiness to receive a third dosage. There were three sections to the questionnaire: (1) demographic data, (2) questions on whether or not respondents felt they needed a booster dosage, and (3) comments about the usefulness of the online resources that were accessed in the course of COVID-19. Age, gender, education, occupation, and place of residence were all examples of sociodemographic questions asked in the first group. Education level was categorized as high-school graduate, college graduate, graduate-school graduate, and non-graduate. There were four age categories defined: those aged 18 to 24, those aged 25 to 34, those aged 35 to 44, and those aged 45 and over. The second portion included a series of yes/no questions about the respondent’s knowledge of and willingness to receive the booster dosage. Finally, we added several questions on the credibility of health-related content found online.

### 2.3. Statistical Analysis

The data from 631 interviews were analyzed by using SPSS v.21.0. Quantitative information was reported as frequencies and percentages. Knowledge, attitude, and practical application were the three types of questions used. In total, 10 of the 22 questions tested general knowledge, and a perfect score was achieved by answering all of them correctly. The correct answer was scored 1, and the wrong answer was scored 0. A person’s knowledge score was calculated by adding up their test scores. Four questions focused on respondents’ attitudes and were assessed in the same way. The attitude questions were scored between 1 to 4 for satisfied, highly satisfied, dissatisfied, and highly dissatisfied. Each individual’s practice score was determined by adding up their scores from 5 practice questions. The answers to three questions did not fit neatly into any of the available options. Thus, it is expected that those having higher knowledge and practice scores would have a lower positive attitude score. Cronbach’s alpha = 0.739 and McDonald’s = 0.754 both indicate that the items in the questionnaire are reliable.

Chi-square and Fischer’s exact tests, with a 95% confidence interval, and a threshold of *p* < 0.05, were used to analyze the significance of the associations between the variables. We conducted one-way analysis of variance to look for correlations between ratings and open-ended queries about our readers’ opinions. The degree of similarity between the three ratings was calculated by using a correlation matrix.

An electronic, anonymized, self-administered questionnaire was created by utilizing a protected survey platform and registered with the number MUREC-LAN.9/COM-2022 I L9-2 after the Institutional Review Board gave their approval for the project. All respondents were made aware that they could leave the study at any moment without any repercussions, and that their participation was entirely voluntary.

## 3. Results

Participant socioeconomic status and willingness to receive a booster dose of the COVID-19 vaccination are shown in Figure 1. The vast majority of the 347 individuals who reported being open to receiving the vaccination were women (319, 91.9%), whereas only 28 (8.1%) were men.

A total of 217 participants (62.5% of the total) was in the 18–24 age range. In spite of this, 179 (63.0%) of these people responded with a negative answer. In addition, 232 (66.0%) of those who had at least graduated high school said they were okay with the extra dosage. When asked the same question, 198 people (or 69.7%) said they were not. More than two-thirds (216) of the Majmaah participants admitted to being vaccinated, and more than half (62%) said they would not be interested in a booster dose.

The distribution of research participants by knowledge, attitude, and practice items is shown in Table 1. Eighty-seven percent of respondents said that they ensured that a source was trustworthy before relying on its claims. About 63.3% of the respondents have access to the information through the Sehaty application, and 14.9% were informed about the COVID-19 vaccine through toll-free lines to ask questions concerning COVID-19. Three-fifths of respondents (59.5%) were worried about the effectiveness of the vaccine and its potential side effects after a third dosage. Of those surveyed, 51.98% said they were familiar with the potential adverse effects, and 72.03% understood the significance of a booster dosage. There were 631 participants, and 517 of them felt they had enough data to make an informed decision regarding COVID-19.

There were two doses of the COVID-19 vaccine given to 80% of the population, with an additional booster vaccination given to 39%. On the other hand, only 54.8% of those who had not yet received the booster vaccine intended to do so. In total, 44.85% were prepared to strike out on their own, if given the opportunity.

About 25.8% of the respondents who were aware of the side effects of the booster dose were planning to receive the COVID-19 booster dose, whereas only few (7%) had not planned to receive the booster dose and were not aware of the side effects after the booster dose. There was a statistically significant (*p* < 0.001) correlation between awareness of the capacity of the booster dose vaccine to prevent problems and willingness to receive a third dose. History of COVID-19 infection was found to be significantly associated with the decisions of the participants to receive the booster dose (*p* < 0.001) (Table 2).

The participants had a knowledge average of 12.9 (+2.61), with a range of 6 to 18. The average score for attitude was 3 (1.2), with a possible range of 0 to 6, while the average score for practice was 4.50 (1.12) (2 to 7). Attitude (*p* < 0.001) and behavior (*p* < 0.001) ratings were substantially correlated with prior COVID-19 immunization status (*p* < 0.005). Those who were vaccinated had a somewhat higher level of understanding about the importance of vaccination (mean score 5.34 (1.09)) than those who were not (mean score 4.36 (1.29)) (*p* < 0.197). Similarly, people who received the vaccine had a significantly higher mean practice score (4.52 (1.12)). Those who had previously received vaccinations had a more favorable outlook on the practice (mean score 3.62 (1.51)) than those who had not (mean score 3.36 (0.809)) (*p* < 0.005). Participants who reported learning about COVID-19 immunization from the Ministry of Health had a higher and statistically significant knowledge score (5.47 (1.02)) (*p* < 0.001). The vaccination-related attitudes (*p* = 0.059) and behaviors (*p* = 0.114) were, however, consistent across all of the research methods (Table 3).

Correlations between the three categories of scores (knowledge, attitude, and behavior) are shown in Table 4. As a matter of statistical significance, r = −0.417 was found between attitude and performance. Since the positive attitude toward vaccination was scored lower than the negative attitude, a negative correlation was observed between attitude and performance score. Equally modest was the positive correlation between knowledge and practice scores, with a value of 0.160 (*p* < 0.001). There was a positive but weak relationship between knowledge and attitude ratings (r = 0.023, *p* = 0.557). This table implies that a person with a positive attitude toward vaccination was significantly more likely to be vaccinated.

## 4. Discussion

Socio-demographic factors were examined in relation to 631 people’s responses on their willingness to receive a third (booster) dose of the COVID-19 vaccine. The majority of individuals were willing to receive the booster dosage (*n* = 347), whereas 284 declined. These results were consistent with those found by Abdulrahman Alamri et al., who found that 2227 individuals in Saudi Arabia were open to receiving a COVID-19 immunization (60%) [10]. Nevertheless, in Bosnia and Herzegovina, researchers found that only 25.7% were interested in receiving the COVID-19 vaccine, while 74.3% were either unsure about being vaccinated or completely refused to do so [11]. In contrast to the findings of a study by EL-Elimat et al., which found that males were more inclined to accept the vaccination than females, the majority of the participants who stated an acceptance of a third dosage of the vaccine were females (91.9%) [12].

We found that our youngest responders, those between the ages of 18 and 24, were the most open-minded. Similarly, El-Elimat et al. found that younger Jordanians were more receptive than those aged >35 [12]. By contrast, the elderly were more receptive to receiving the third dosage of COVID-19 in a study by AL-Mohaithef et al [4].

According to the current research, people with a bachelor’s or diploma degree were more likely to accept the third dosage (66.9%) than those with a higher level of education. Contrary to this, Eman Ibrahim Alfageeh et al. [13] found that 52% of individuals with a higher education level declined vaccination. The majority of vaccination accepters (67.1%) were unmarried people, whereas those with partners (30.6%) were more likely to say no. Previous COVID-19 infection was positively associated with accepting the booster dosage, as shown by our findings. Our findings showed that 114 people with a prior history of COVID-19 infection (40.3% of the total) declined to receive the booster dose. A larger percentage, 257 (73.9%), of those who had never been infected said they planned to be vaccinated. The rejection rate for the COVID-19 booster vaccination was low among the Saudi population, as shown by this survey of 631 people. While 346 (54.8%) of the participants were aware of the potential negative effects from the booster dosage, this did not dampen their enthusiasm for the treatment. Nevertheless, 163 (25.6%) of respondents who worried about the potential negative consequences of the booster dose said that they still intended to be vaccinated. Yet another study, this one published by Altulahi et al., found that concerns about possible adverse reactions to the COVID-19 vaccine were the primary factor contributing to people’s decision not to be vaccinated [14]. However, among those who took part in our research, 54.8% were committed to receiving the COVID-19 booster dose.

This research highlights the significance of the Ministry of Health communications in promoting the uptake of the general public of the COVID-19 vaccination. Forcareli et al., who did their research in Naples, Italy, found results that were quite similar [15]. According to their findings, people who accessed the material on the official website of the COVID-19 vaccine were 1.68 times more likely to agree to receive a booster dose (*p* < 0.034).

Awareness of the efficacy of the vaccine, confidence in the potential of the vaccine to prevent problems, and acceptance of a third dosage are all significantly related, as the results demonstrate. These findings are consistent with those of recent Saudi research conducted by Mubarak et al. [16] which found that among students at Taif University, there was a substantial connection between vaccination intent and optimistic views of vaccine safety and effectiveness.

About 70% of all participants relied on the Ministry of Health as their primary source for acquiring health information during a crisis, with social media coming in at a distant second with 24.2% of all participants. Alshareef et al. found, in contrast, that the general public in Saudi Arabia relied mostly on social media platforms to learn about COVID-19, rather than the Ministry of Health, which ranked only second [17]. Several of our participants expressed extreme satisfaction with the quality of online COVID-19 health resources. It has been suggested that the content should be tailored to advocate the importance of vaccination, its efficacy, and safety. It should also focus on clarification of misinformation and information pertaining to health policies [18].

### Strength and Limitation

This study adds to the growing body of literature on the acceptance of booster doses of the COVID vaccine and the role of online resources in increasing its compliance. Given that this study is cross-sectional and involved an online survey conducted through social media platforms, it is subjected to recall bias as well as social acceptability. These factors act as limitation for the generalization of the results on the general population as a whole. Another limitation is that the non-response rate cannot be deduced even though the study caters to a large population.

## 5. Conclusions

According to our findings, past COVID-19 infection is positively associated with acceptance of a booster dosage, and females, single persons, those between 18 and 24 years old, and those with a bachelor’s or certificate degree are more likely to receive the third dose of the vaccine. Moreover, the survey demonstrated that many individuals look to the health ministry as their primary resource for learning about COVID-19 health issues. There was a correlation between vaccination knowledge, confidence in the capacity of the vaccine to prevent problems, and willingness to receive a third dose. In spite of the government’s efforts to reach out to encourage the general population to receive the booster dose, nearly 40% of the participants were not willing to receive the booster dose. Hence, creating an educational framework that can educate the community on the dangers of not being vaccinated or waiting would be a good start. In particular, combating the disease would benefit greatly from an open educational effort that emphasizes the societal advantages of receiving the vaccine.

## Figures and Tables

**Figure 1 vaccines-11-00929-f001:**
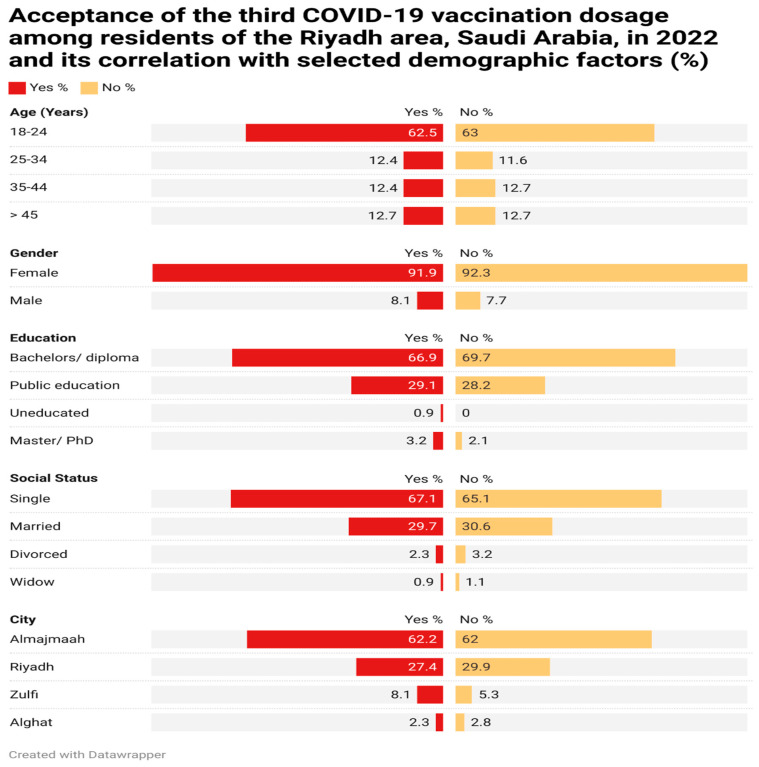
Acceptance of the third COVID-19 vaccination dosage with regards to selected demographic factors.

**Table 1 vaccines-11-00929-t001:** Distribution of the study participants based on items in the questionnaire [N = 631].

Knowledge-Based Items	n (%)	95% Confidence Interval	
Whether the reliability is checked of the source before up-taking the information	553 (87.64%)	84.8%	90.11%
Knowledge about the application of sehaty	602 (95.40%)	93.47%	96.90%
Knowledge about the toll-free number (937) services of the Saudi Ministry of Health to inquire about COVID-19	421 (66.72%)	62.89%	70.39%
Aware of the importance of the third dose	450 (71.32%)	67.61%	74.82%
Aware of any side effects of the third dose	328 (51.98%)	48.00%	55.94%
Concerned about the efficacy and duration of the third dose of the vaccine	376 (59.59%)	55.64%	63.44%
**Attitude-based items**			
Proportion worried about becoming infected with COVID-19 even after becoming the vaccination	296 (46.91%)	42.96%	50.89%
Proportion satisfied with the amount of health information available about COVID-19	517 (81.93%)	78.71%	84.86%
Receiving the third dose (booster) of the COVID-19 vaccine decreases my chance of getting COVID-19 or its complications	182 (28.84%)	25.34%	32.55%
Planning to receive the third dose	346 (54.83%)	50.86%	58.77%
Would you prefer to take a booster dose if given a choice?	283 (44.85%)	40.92%	48.83%
Would receive the third dose only if it becomes mandatory by MOH	439 (69.57%)	65.82%	73.14%
**Practice-based items**			
Vaccinated against COVID-19	620 (98.26%)	96.90%	99.13%
No. of Doses received			
1 dose	11 (1.74 %)	0.87%	3.10%
2 doses	387 (61.33%)	57.41%	65.15%
3 doses	233 (36.93%)	33.15%	40.82%
Usage of Sehaty to obtain information about COVID-19	400 (63.39%)	59.50%	67.16%
Usage of toll-free number	94 (14.90%)	12.21%	17.92%
Vaccinated against COVID-19, booster	233(36.93%)	33.15%	40.82%

**Table 2 vaccines-11-00929-t002:** Association between acceptance of COVID-19 booster dose and awareness and practice-based questions.

Awareness-Based Questions		Planning to Take a COVID-19 Booster Dose			*p*-Value
		Yes	No	I Do Not Know	
Awareness of any side effects of the booster dose	Yes	163 (25.8%)	63 (10%)	102 (16.2%)	0.025
	No	183 (29%)	44 (7%)	76 (12%)	
Receiving the third dose (booster) of the COVID-19 vaccine decreases the chance of contracting COVID-19 or its complications.	Disagree	44 (7.0%)	47 (7.4%)	20 (3.2%)	<0.001
	Strongly disagree	30 (4.8%)	30 (4.8%)	11 (1.7%)	
	Agree	28 (4.4%)	182 (28.8%)	85 (13.5%)	
	Strongly agree	5 (0.8%)	87 (13.8%)	62 (9.8%)	
Practise-based questions		Acceptance of COVID-19 Booster			*p*-value
		Yes	No	<0.001	
History of infection with COVID-19	Yes	91 (26.1%)	114 (40.3%)		
	No	257 (73.9%)	169 (59.7%)		

**Table 3 vaccines-11-00929-t003:** Association between knowledge, attitude, and practice score and history of vaccination, source of information, and opinion about it among participants [N = 631].

Predictors		N	Knowledge Score		F-Test (df)	*p*-Value	Attitude Score		*t*-Test (df)	*p*-Value	Practise Score		*t*-Test (df)	*p*-Value
			Mean	SD			Mean	SD			Mean	SD		
History of vaccination	Unvaccinated		4.36	1.286	−1.29 (629)	0.197	3	1.414	−3.4 (629)	<0.001	3.36	0.809	−2.79 (629)	0.005
	Vaccinated		5.34	1.09			3.62	1.51			4.52	1.12		
Source of Information	Ministry of Health	443	5.47	1.021	7.8 * (3, 21.2)	<0.001	3.49	1.54	2.89 * (3, 21.6)	0.059	4.56	1.12	2.227 * (3, 21.4)	0.014 #
	Social Media	169	5.05	1.182			3.87	1.41			4.3	1.13		
	Relatives and friends	10	4.4	1.35			4.1	1.73			4.5	1.08		
	others	9	4.44	1.509			3.78	1.2			4.78	1.3		
Opinion about the information	Satisfied	251	5.21	1.13	3.46 * (3, 79.3)	0.02	3.86	1.46	23.7 * (3, 82.7)	<0.001	4.33	1.069	11.78 * (3, 84)	<0.001 #
	Very satisfied	317	5.46	1.04			3.2	1.45			4.72	1.155		
	Unsatisfied	34	5	1.28			4.62	1.35			3.97	0.834		
	Very Unsatisfied	29	5.24	1.15			4.66	1.23			4.07	1.033		

* F value- significant, # *p* value- significant.

**Table 4 vaccines-11-00929-t004:** Correlation matrix between knowledge, attitude, and practice score [N = 631].

		Practice Score	Knowledge Score
**Knowledge score**	Pearson’s r *p*-value	0.160 *** <0.001	
**Attitude score**	Pearson’s r *p*-value	−0.417 *** <0.001	0.023 0.557

*** *p* < 0.001.

## Data Availability

The data will be available with the corresponding author as and when requested.

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
