# Peer review of "Awareness and Acceptance for COVID-19 Booster Dose Vaccination among Residents of Saudi Arabia: Findings of a Cross-Sectional Study"

_vaccines, 2023, doi:10.3390/vaccines11050929_

Round 1
Reviewer 1 Report
General Comment
This is an important study investigating the acceptance of the COVID-19 booster dose vaccination among residents in KSA.
Major comments
Revise the title of the manuscript to include the study design.
Give more details of the recruitment of participants. It’s representative of the entire KSA.
Why not a survey in Arabic and English? (KSA have more than 6 million of foreign people).
Several references don’t match (e.g., Line 256 the reference is ‘17’ not ‘16’)
Minor comments
There are some English grammar errors which should be looked at. I recommend authors work with someone to check for these errors (e.g., line 27,90)
How many people were invited to take the survey? What was the response level?
I recommend structuring better materials and methods (e.g., setting and participants; procedures; Questionnaire; statistical analysis)
Can you add a section in the manuscript with ‘strengths and limitations of this study’?

Author Response
Major comments
Comment 1: Revise the title of the manuscript to include the study design.
Response: We sincerely thank the reviewer for raising this important issue. The study design is included as suggested.
Comment 2: Give more details of the recruitment of participants. It’s representative of the entire KSA.
Response: We sincerely thank the learned reviewer for the positive comments and the constructive feedback that has been provided to improve our article. The research team which was responsible for floating the questionnaire link, belonged to four different Riyadh Region of Saudi Arabia (including the cities of Riyadh, Al Majma'ah, Al Ghat, and Zulfi). They used their social media handles to cater to larger representative population.
Comment 3: Why not a survey in Arabic and English? (KSA have more than 6 million of foreign people).
Response: Thanks for pointing out this. Since the survey was limited to residents of KSA, only Arabic language was used in the questionnaire.
Comment 4: Several references don’t match (e.g., Line 256 the reference is ‘17’ not ‘16’)
Response: The references are rechecked and corrected.
Minor comments
Comment 5:There are some English grammar errors which should be looked at. I recommend authors work with someone to check for these errors (e.g., line 27,90)
Response: Thank you for your suggestion. I have tried to correct it as much as possible.
Comment 6:How many people were invited to take the survey? What was the response level?
Response: This is very difficult to calculate as different social media platform was used to invite the participants.
Comment 7: I recommend structuring better materials and methods (e.g., setting and participants; procedures; Questionnaire; statistical analysis)
Response: Thank you for your valuable suggestion. I have structured the methodology.
Comment 8: Can you add a section in the manuscript with ‘strengths and limitations of this study’?
Response: Thanks to the learned reviewer for providing the very detail revision of our article and providing excellent suggestions for its improvement. I have added this section.
Reviewer 2 Report
The purpose of this study, according to the authors, was to evaluate the quality of web-based information gathered. during COVID19 in Saud Arabia and determine the extent to which participants in the survey were aware of the need for a booster dosage. To gather data, the authors conducted a COVID19 cross-sectional online survey that requested demographic data; questions on whether respondents felt they needed a booster dosage; and comments on how online resources influenced decisions to seek a booster dose.
The sample size of 631 respondents was sufficient for statistical analysis. Authors could you provide a more detailed explanation of how you derived the data values in Table 3? Also what is the significance of the comparison in Table 4, not sure the reader will understand this data?
The survey results are not surprising and agree with similar publications from surveys done in other countries and in Saud Arabia. Authors in your Discussion the last sentence in lines 271-272 is rather negative " prior research has cautioned that online health information may be erroneous and this might have serious consequences". Your data has more positive spin than this statement. Yes surveys may lead to some erroneous conclusions, but why not revamp the concluding statement to a more positive outcome based on your data?
Author Response
Comment: The sample size of 631 respondents was sufficient for statistical analysis. Authors could you provide a more detailed explanation of how you derived the data values in Table 3? Also what is the significance of the comparison in Table 4, not sure the reader will understand this data?
Response: Thanks to the learned reviewer for providing the very detail revision of our article and providing excellent suggestions for its improvement. The scoring used for the calculation of knowledge, attitude and practice score is detailed in the methodology section. The significance of comparison is that, those having better knowledge had positive attitude and good practice score as well. This is further elaborated in the result section as suggested.
Comment: The survey results are not surprising and agree with similar publications from surveys done in other countries and in Saud Arabia. Authors in your Discussion the last sentence in lines 271-272 is rather negative " prior research has cautioned that online health information may be erroneous and this might have serious consequences". Your data has more positive spin than this statement. Yes surveys may lead to some erroneous conclusions, but why not revamp the concluding statement to a more positive outcome based on your data?
Response: Thanks to the learned reviewer for providing important suggestions that can improve the quality of manuscript. We have changed the concluding lines accordingly.
Reviewer 3 Report
Very interesting topic.
Poor introduction, not sound methodology (not random sampling, selection bias). Serious problems with result presentation (e.g. no condidence intervals).
No mention of numerous study limitations.
Conclusions are not supported by the results.
Author Response
Comment: Poor introduction, not sound methodology (not random sampling, selection bias). Serious problems with result presentation.
Response: We sincerely thank the learned reviewer for the comment. The introduction section is rewritten and enhanced. The study being online, study design and various bias has been written as part of limitation of study. The mean and SD have been depicted in table 3 and excplaantion for the table 3 and 4 has been rewritten.
Comment: No mention of numerous study limitations.
Response: Thanks for pointing out this. The study limitations have been added in detail at the end of the discussion section.
Comment: Conclusions are not supported by the results.
Response: We sincerely thank the learned reviewer for the positive comments and the constructive feedback that has been provided to improve our article. The conclusion part is rewritten to suit the study results and has been enhanced.
Round 2
Reviewer 3 Report
Thank you for addressing most of my suggestions. Your manuscript is seriously improved.
Please add 95% CI's at lest to your major findings since you are using a sample and you should depict the random error of your findings.
Your Abstract needs carefull attention and some corrections.
Background: In the global effort to combat coronavirus-19 you mean SARS-CoV-2, adoption of the vaccination has been an essential component. The goal of this research was to determine whether or whether ?? residents of the Riyadh metropolitan area were ready to submit to a third COVID-19 vaccination dosage and, if so, how soon they would be willing to do so; Methods: To gauge interest in and support for a booster shot, as well as satisfaction with the availability and accuracy of internet resources, researchers performed a cross-sectional online survey. Total replies from 631 individuals. All adult men and females in the Riyadh area of Saudi Arabia became eligible for participation. This study included people from the cities of Riyadh, Al Majma'ah, Al Ghat, and Zulfi in the Riyadh Area; Results: Knowledge of the vaccine's efficacy, confidence in the vaccine's capacity to avert problems, and willingness to get a third dosage were all shown to be substantially correlated (p<0.001). De-spite widespread fear of the potential harmful consequences of the COVID-19 booster, about 54.8% of individuals nevertheless agreed to get it; Conclusions: Females, those without children, those aged 18-24, and those with a bachelor's or di-ploma were more likely to take a third dose of the COVID-19 vaccination, and there was a signif-icant correlation between prior COVID-19 infection and vaccine acceptance.All these are results not conclusions Findings indicate that the health ministry is a primary entry point for many in the search for information regarding COVID-19 in the context of health.
Author Response
We are pleased with the feedback you have provided in regard to considering that the paper has potential and for wishing us the courage and good luck for the further process. We have taken all your recommendations on board and made the necessary modifications and improvements in the revised version of our paper.
Comment : Please add 95% CI's at lest to your major findings since you are using a sample and you should depict the random error of your findings.
Response: Thanks for pointing out this. As suggested the confidence intervals have been added in table 2.
Comment: Your Abstract needs careful attention and some corrections.
Response: We sincerely thank the learned reviewer for the positive comments and the constructive feedback that has been provided to improve our article. The abstract is rewritten as a whole. Thanks for pointing out the mistakes in the abstract. Now the abstract is much more improved and looks good.